# Prediction of Drug-Target Affinity Using Attention Neural Network

**DOI:** 10.3390/ijms25105126

**Published:** 2024-05-08

**Authors:** Xin Tang, Xiujuan Lei, Yuchen Zhang

**Affiliations:** 1School of Computer Science, Shaanxi Normal University, Xi’an 710119, China; 2College of Information Engineering, Northwest A&F University, Xianyang 712199, China; yczhang@nwafu.edu.cn

**Keywords:** drug-target interactions (DTI), deep learning, Bidirectional Gated Recurrent Unit (BiGRU), Graph Sample and Aggregate (GraphSAGE), attention neural network (ANN)

## Abstract

Studying drug-target interactions (DTIs) is the foundational and crucial phase in drug discovery. Biochemical experiments, while being the most reliable method for determining drug-target affinity (DTA), are time-consuming and costly, making it challenging to meet the current demands for swift and efficient drug development. Consequently, computational DTA prediction methods have emerged as indispensable tools for this research. In this article, we propose a novel deep learning algorithm named GRA-DTA, for DTA prediction. Specifically, we introduce Bidirectional Gated Recurrent Unit (BiGRU) combined with a soft attention mechanism to learn target representations. We employ Graph Sample and Aggregate (GraphSAGE) to learn drug representation, especially to distinguish the different features of drug and target representations and their dimensional contributions. We merge drug and target representations by an attention neural network (ANN) to learn drug-target pair representations, which are fed into fully connected layers to yield predictive DTA. The experimental results showed that GRA-DTA achieved mean squared error of 0.142 and 0.225 and concordance index reached 0.897 and 0.890 on the benchmark datasets KIBA and Davis, respectively, surpassing the most state-of-the-art DTA prediction algorithms.

## 1. Introduction

Exploring drug-target interactions (DTIs) is instrumental in elucidating the mechanism of drug action, thereby offering valuable insights for drug design and development. However, constrained by manpower, material resources, and financial resources, conventional biological experiments like high-throughput screening struggle to achieve large-scale applications and meet the practical requirements. Consequently, the precise and reliable calculation predictive methods for DTIs have become a prevalent tool in DTI research.

Traditional DTI computational methods encompass molecular dynamics simulation [1] and molecular docking [2]. Molecular docking can emulate the docking modes of protein macromolecules and small compound molecules, thereby simulating a variety of potential binding poses. It then calculates scoring functions to minimize the free energy at binding sites. Despite having strong biological interpretability, molecular docking requires substantial computing resources and exhibits slow calculation speeds. Furthermore, the limited availability of proteins with accurate 3D crystal structures restricts the applicability of these algorithms.

In recent years, the emergence of publicly available drug and target databases has highlighted the potential of machine learning (ML). As a data-driven computational method, ML effectively leverages vast amounts of data from related databases for supervised learning, thereby expanding its application prospects in drug discovery. Initial research treated DTI prediction as a binary classification task, solely distinguishing between combination and non-combination categories. Inspired by the methods employed in other association prediction studies in the field of bioinformatics [3,4,5,6,7,8], these methods incorporated pharmacological data of drugs and targets or constructed heterogeneous networks that linked drugs, targets, and other biological entities for prediction [9]. However, these methods incur some degree of information loss, as well as challenges including determining the threshold for combination and non-combination, and the lack of reliable non-combination samples [10].

Subsequently, Tang et al. [11] proposed modeling DTI predictions as a regression task using drug-target affinity (DTA) to precisely reflect the DTI intensity. DTA is a category of data capable of illustrating the strength of the binding interaction between drugs and targets. Typically, this refers to the dissociation constant (*K_d_*), inhibition constant (*K_i_*), and half-maximal inhibitory concentration (*IC*_50_). The lower these values are, the greater the affinity.

Early ML-based DTA prediction methods predominantly employed traditional ML techniques. Example of this are the Kronecker regularized least-squares (KronRLS) algorithm for DTA prediction proposed by Tang et al. [11] and the gradient boosting-based DTA prediction algorithm SimBoost [12] developed by He et al.

These methods heavily depend on intricate feature engineering and necessitate extensive expert domain knowledge. Additionally, the features extracted artificially often encounter issues such as information loss and an inability to adapt to specific tasks. In contrast, deep learning (DL) methods integrate feature representation learning and model training within an end-to-end architecture, which allows for the automatic learning of effective representations from raw drug target data, capturing potential rules of DTAs. Consequently, these methods exhibit superior generalization capabilities on larger datasets and have achieved significant enhancements in prediction accuracy.

Representing both drug and target as 1D sequences (i.e., the simplified molecular-input line-entry system (SMILES) of drugs and amino acid sequences of target proteins), commonly used DL algorithms in natural language processing (NLP) was employed in DTA prediction, such as Convolutional Neural Networks [13,14,15] (CNN), Recurrent Neural Networks [16,17] (RNN), and transformer [18,19]. The potential of attention mechanisms was also explored [20,21]. Theoretically, these algorithms can extract DTA-related features automatically from the raw full target residue sequence in the absence of protein binding pocket structure. DeepDTA [13] proposed by Öztürk et al. is the earliest DL-based DTA prediction algorithm, which adopts CNN to learn representations from drug SMILES and amino acid sequences of target protein separately, then concatenates them and predicts DTAs through a fully connected network. Zhao et al. propose AttentionDTA [20], which introduces two attention mechanisms based on DeepDTA to focus on the important parts of protein (drug) sequences according to drug (protein) sequences. MT-DTI [19] proposed by Shin et al. utilizes a multilayer bidirectional transformer to encode drug SMILES to capture long-distance relationships between atoms in drugs. MGPLI [18] proposed by Wang et al. uses both character-level and fragment-level features and adopts a transformer-CNN encoder to extract high-level drug and target features followed by highway feedforward layers to solve feature redundancy.

Drugs can also be represented as 2D molecular graphs, which is advantageous for capturing the topological structures of drug molecules. Consequently, numerous researchers have integrated encoders designed for sequence and Graph Neural Networks (GNN), developing various DTA prediction algorithms. Nguyen et al. proposed GraphDTA [22], which employed the open-source cheminformatics software RDKit (version 2023.3.3) [23] to transform SMILES of drugs into molecular graphs. Subsequently, they utilized four types of GNN: GCN (Graph Convolutional Network), GAT (Graph Attention Network), GIN (Graph Isomorphism Network), and a combined GAT-GCN architecture to generate drug representations. Lin introduced DeepGS [24], which employs Smi2Vec and Prot2Vec to encode target sequences respectively. Subsequently, it utilizes GAT and Bidirectional Gated Recurrent Unit [25] (BiGRU) to learn the multimodal representations of drugs. Both GraphDTA and DeepGS still use CNN to learn target representations.

In this study, we proposed a novel algorithm based on Graph Sample and Aggregate (GraphSAGE) [26], BiGRU, and Attention Neural Network (ANN) for DTA prediction, named GRA-DTA. Target protein sequence representations are learned using an attention-based BiGRU which can maintain context, thereby learning the sequence representation of target proteins and emphasizing crucial sections related to DTA in extended protein sequences. Meanwhile, to harness the topological information of drugs, drug molecules are modeled as graphs, of which representations are learned via GraphSAGE. Subsequently, an ANN is applied to capture the varying attention weights of each specific drug-target pair by merging drug and target representations and then obtaining the representations. Finally, representations of drug-target pairs are fed into fully connected layers and continuous values of unknown DTAs are predicted. Experiments on domain benchmark datasets demonstrate that GRA-DTA exhibits high accuracy and surpasses baseline methods. The contributions of this article are delineated below:(1)We propose a new DTA prediction algorithm, GRA-DTA, based on soft attention-based BiGRU, GraphSAGE, and ANN.(2)We conduct comparative experiments on benchmark datasets to substantiate the efficacy of the proposed algorithm. Additionally, we conduct ablation experiments to demonstrate the significance of individual modules. Furthermore, we assess the performance of GRA-DTA across three experimental scenarios of cold start. The case study focused on the COVID-19 target proves the application prospect of GRA-DTA in drug repurposing.

## 2. Results

### 2.1. Experimental Setting

We introduce Davis and KIBA for evaluation of GRA-DTA. In the experiment, the datasets were first partitioned into two segments by a ratio of 5:1 for training and independent testing.

We deployed the experiment on the NVIDIA 3090 with 8 GB memory. The optimizer was Adam optimizer with a learning rate set to 2 × 10^−4^. For the smaller Davis datasets, the batch size was set to 128. In the case of the larger KIBA datasets, the batch size was 256. The number of training epochs was set to 600, and a dropout rate of 0.2 was applied.

Several key hyperparameters, including the layer of in BiGRU and the layer of GraphSAGE, determine the model structure and thus impact the overall performance. To identify the optimal parameter settings, five-fold cross-validation (5-CV) was conducted using the smaller Davis dataset. Specifically, the training set was further randomly split into five folds of equal size, with each fold alternately used as the validation set, and the remaining four folds used as the training set. The average of the five-fold results served as final performance for a particular hyperparameter combination.

The specific hyperparameter settings are illustrated in Table 1. 

### 2.2. Comparison with Other Algorithms

To showcase the effectiveness and advancement of GRA-DTA, we have chosen five state-of-the-art DL algorithms as baselines: DeepDTA, MT-DTA, DeepGS, GraphDTA, and MGPLI. For a fair comparison, baseline algorithms followed the same training–testing split methodology as GRA-DTA. We evaluated the performance of algorithms with the mean squared error (*MSE)*, concordance index (*CI*), and the regression toward the mean index ($r_{m}^{2}$), whose calculation formulas are given in Section 4.2. The comparative results are presented in Table 2 and Figure 1.

On the Davis dataset, GRA-DTA achieves the second lowest *MSE*, surpassed only by MGPLI. For *CI* $r_{m}^{2}$, it achieves a performance improvement of 0.01 and 0.029 compared to the suboptimal methods. On the KIBA dataset, GRA-DTA achieves a performance improvement of 0.017 and 0.031 on *MSE* and $r_{m}^{2}$, respectively, while *CI* is only 0.01 lower than MGPLI.

### 2.3. Ablation Experiments

To ascertain the efficacy of each component of GRA-DTA and to discern the primary factors that impact performance, we undertook an ablation study.

In this section, GraphDTA is employed as our baseline. In contrast to GraphDTA, GRA-DTA replaces the CNN encoder for target protein sequences with soft attention-based BiGRU, uses GraphSAGE instead of GIN as drug molecule graph encoder, and learns drug-target pair representation by ANN rather than simple concatenation. We conduct ablation experiments by gradually removing components of GRA-DTA. The following are variants of our algorithm:

GRA_no_att: remove the soft attention module from the original framework and directly input the flattened BiGRU output into a linear layer to derive the representation of the target protein.

GRA_no_ann: remove the ANN module from the original framework and simply concatenate the representations of drugs and targets to predict DTA.

GRA_no_att_ann: remove both the ANN module and soft attention module from the original framework.

The results of the ablation experiments are presented in Table 3 and Figure 2.

In general, the overall trend of experimental results on Davis and KIBA datasets, as depicted in Figure 2, indicates that the removal of any individual component leads to a degradation in prediction performance. Notably, the full GRA-DTA delivers superior predictive performance.

The GRA_no_att_ann outperforms Graph-DTA, which is ascribed to the capacity of BiGRU to capture the context dependency of protein long sequences more effectively than CNN, which solely learns local features of sequences. The GRA_no_ann with soft attention decreases the *MSE* by 0.002 and increases the *CI* by 0.008 on the Davis dataset, decreases the *MSE* by 0.006, and increases the *CI* by 0.002 on the KIBA dataset, which is because the attention mechanism focused on important parts of DTA in long sequences of proteins. The GRA_no_att with ANN decreases the *MSE* by 0.03 and increases the *CI* by 0.007 on the Davis dataset, decreases the *MSE* by 0.08, and increases the *CI* by 0.001 on the KIBA dataset. The reason for the performance improvement is that the ANN considers different contributions of different features and dimensions.

### 2.4. Performance Evaluation on Cold Start Scenarios

Prior experiments randomly split the training and testing sets, resulting in an overlap of drugs and targets between testing and training sets. However, DTA prediction algorithms are typically employed for screening novel candidate compounds and target discovery in real-world drug discovery. The algorithms must predict the affinity of drugs (targets) without any known affinity, which implies that there is no overlap between the drugs or targets in the testing set and the training set. To accomplish this task, the algorithm must possess robust generalization capabilities to discover potential patterns in DTI. We establish three cold-start scenarios to evaluate the efficacy of the GRA-DTA in practical application scenarios.

Cold drug scenario: the drugs present in the training dataset are absent from both the validation and testing sets.

Cold Target scenario: the targets present in the training dataset are absent from both the validation and testing sets.

Cold drugs-target scenario: both the drugs and targets present in the training set are absent from validation and testing sets.

In this section, we select GraphDTA and MGPLI, which exhibit optimal performance under a random partition as our baselines. In each experiment, the training set, validation set, and testing set are split by a ratio of 8:1:1. To ensure the stability of our experimental results, we repeat the experiment five times, averaging the values to obtain the final result and employ the same method for dividing the training-testing sets across all methods. The experimental results of all methods under cold start scenarios are presented in Table 4 and Figure 3.

GRA-DTA outperforms in all scenarios except for the cold-drug scenario on the Davis dataset. This discrepancy may be caused by the limited number of unique drugs (68), which may lead to insufficient model training. However, under cold target and cold drug-target scenarios, our algorithm shows an overall improvement of 10.7% and 13.5% in *MSE* and *CI* compared to suboptimal methods.

The experimental results on the KIBA dataset are more representative, with GRA-DTA showing an overall improvement of 4.7%, 11.1%, and 4.1% in *MSE* and *CI* compared to suboptimal methods under cold-drug, cold-target, and cold target-drug scenarios. The significant performance under the cold-target scenario suggests that our algorithm is more effective at capturing protein features.

### 2.5. Case Study

We also undertake a case study focused on COVID-19 to further assess the efficacy of GRA-DTA in practical drug repurposing. We chose SARS-CoV-2 3C-like protease [27] (3CLpro) as the target. This cysteine protease is crucial in genome replication and the expression of coronaviruses, and it has emerged as a significant target for drug development and antiviral research.

We chose 84 antiviral drugs that have been approved for marketing and 3 unrelated drugs, namely Artemisinin, Penicillin, and Aspirin. SMILE strings of the 87 drugs obtained from PubChem were combined with the amino acid sequences of 3CLpro obtained from Uniprot to form drug-target pairs, which were input into the GRA-DTA trained based on the larger KIBA dataset.

Table 5 provides a partial view of the prediction results. Among the top 10 drugs predicted by our GRA-DTA with the highest affinity to 3CLpro, 6 have been confirmed to have a certain therapeutic effect on COVID-19 by relevant literature research.

Ribavirin, with top-ranked predicted affinity, is a guanosine analog that disrupts the replication of RNA and DNA viruses. The second-ranked Didanosine is an inosine/adenosine/guanosine analog, both of which were initially used to treat infection of human immunodeficiency virus (HIV). According to the Fifth Edition of the Treatment Protocol, the Chinese government has recommended the use of Ribavirin for the treatment of COVID-19 [28]. There are also studies proving that the median effective concentration (EC_50_) value of Didanosine against SARS-CoV-2 in vitro exceeds that of Remdesivir, which has been approved for the treatment of COVID-19 [32]. The 3D pose of the ligand-protein binding state between the two drugs, including Ribavirin (PubChem CID: 37542) and Didanosine (PubChem CID: 135398739) with 3CLpro (PDB ID: 7NXH), is plotted in Figure 4.

The ranks of 3 unrelated drugs we added are 67, 79, and 87 among the 87 drugs, which is consistent with reality and proves the reliability of our algorithm in drug repositioning.

## 3. Discussion

In this study, we introduce a novel DL-based algorithm GRA-DTA for DTA prediction, and conducted comparative experiments between GRA-DTA and state-of-the-art DL algorithms on the benchmark datasets Davis and KIBA.

To provide further insight into the performance of GRA-DTA, Figure 5shows the distribution of prediction results for test samples on the Davis dataset and KIBA dataset, where the X-axis and Y-axis represent the predicted values and actual affinity values of identical samples. The red solid line delineates the linear fit to the scatter points, and the black dottedline is *y* = *x* representing perfect prediction. It can be seen that the scatters are densely distributed around the black dashed line on both datasets, and the red solid line coincides with the black dashed line, which indicates that the algorithm has good predictive performance. We also calculated the spearman correlation coefficient of our model, which is 0.712 on Davis dataset and 0.883 on KIBA dataset.

According to the evaluation results presented in Section 2.2, GRA-DTA demonstrates competitive performance against baseline algorithms. This is primarily attributed to the fact that DeepDTA, MT-DTA, and MGPLI only utilize drug SMILES, whereas GRA-DTA employs the molecular graph representation of drugs, thus avoiding the loss of topology information in the molecular structure. Additionally, GraphDTA and DeepGS solely employ a CNN encoder for target protein feature extraction. However, since CNNs can only capture local correlations in sequences and protein sequences are lengthy, a simple CNN fails to adequately capture the contextual dependencies in protein sequences. GRA-DTA utilizes a soft attention-based BiGRU, which is more adept at capturing long sequence features and aggregates the target protein sequence features using a soft attention mechanism to amplify the significance of residue features strongly correlated with DTA in the target protein sequence. Moreover, unlike all baseline algorithms, which simply concatenate the feature vectors of drugs and targets for DTA prediction, GRA-DTA uses an ANN to fuse the representations of drugs and targets, thereby contributing to enhancing prediction performance.

It is also observed that *MSE* on the Davis dataset is always worse than that on the KIBA dataset, which indicates that it is more challenging to obtain an ideal *MSE* on the Davis dataset. This difficulty arises due to the label distribution on this dataset is concentrated around a smaller value of five, and the number of samples with small label values significantly exceeds that of those with large values. Consequently, the algorithm is prone to predict a minor affinity and fall into local optimums. As illustrated in Figure 5, the distribution of dots above and below y = x is not balanced, with more dots being above the y = x. This suggests that more predicted affinity is smaller than actual affinity. Moreover, the KIBA dataset is approximately four times larger than the Davis dataset. Therefore, the KIBA dataset can provide a more comprehensive and diverse data distribution, which enables the model to be trained more effectively, enhancing its generalization capabilities and mitigating the risk of overfitting. So, it showed superior performance in *MSE*, a metric that accurately quantifies the numerical discrepancy between the predicted and actual affinities. In conclusion, to enhance the prediction accuracy on the Davis dataset, well-designed mechanisms are needed to prevent the model from prematurely converging to an unbalanced state. Controlling the complexity of the model to predict overfitting is also important. Furthermore, it also can be seen that the *CI* on the KIBA dataset is marginally lower than that on the Davis dataset for all algorithms, which can be attributed to the labels on the KIBA dataset being characterized by minimal differences and low discrimination. This makes it challenging to promote *CI*, which focuses on assessing the ranking capabilities of algorithms.

Moreover, we identified a few limitations of GRA-DTA during our experiments. According to the evaluation results on cold start scenarios presented in Section 2.3, it can be seen that GRA-DTA falls behind GraphDTA on the Davis dataset in the cold-drug scenario. This discrepancy may be caused by the limited number of unique drugs (68), which may lead to insufficient model training. It also indicates that our proposed algorithm has shortcomings in capturing drug features, which is our future improvement direction. We believe that the inclusion of more additional chemical information of drugs, using multimodal representations of drugs or introducing the utilization of pre-trained models and transfer learning, will help to solve this problem.

## 4. Materials and Methods

### 4.1. Benchmark Datasets

In this study, we selected two commonly used benchmark datasets in DTA prediction: Davis and KIBA.

The Davis dataset was collected by Davis et al. [37] and contains 30,056 binding affinity values between 68 drugs and 442 target proteins, which are represented by *K_d_*. To reduce the range of *K_d_* values, Similar to He et al. [12], we transformed *K_d_* into log space and calculated *pK_d_* as a measure of affinity.
(1)$$pK_{d}=-\log_{10}(\frac{K_{d}}{10^{9}})$$

The KIBA dataset was collected by Tang et al. [38] and then filtered and normalized by He et al. [12]. KIBA consists of 118,254 binding affinity values expressed as KIBA *scores* between 2111 drugs and 229 target proteins. The KIBA *score* integrates the information contained in *K_i_*, *K_d_*_,_ and *IC*_50_, which is calculated as shown in Equation (2) in which *L_i_* and *L_d_* are two fixed weight parameters.
(2)$$\begin{array}{c} K_{i}\prime =\frac{IC_{50}}{1+L_{i}(IC_{50}/K_{i})}\\ K_{d}\prime =\frac{IC_{50}}{1+L_{d}(IC_{50}/K_{d})}\\ \mathrm{KIBA}\;score=\left\{\begin{array}{c} K_{i}\prime \\ K_{d}\prime \\ (K_{i}+K_{d})/2 \end{array}\begin{array}{c} \;if\;IC_{50}\;and\;K_{i}\;are\;present\\ \;if\;IC_{50}\;and\;K_{d}\;are\;present\\ \;if\;\;all\;present \end{array}\right. \end{array}$$

The chemical structure information of the drugs is represented by SMILES, which were collected from the PubChem database [39]. The primary information of target proteins was represented by amino acid sequences, which were collected from the UniProt database. The results of the datasets are summarized in Table 6.

Figure 6 depicts the distribution ranges of affinity values, drug SMILES length, and amino acid sequence length in Davis and KIBA datasets. It shows that the SMILES length of most drugs in the two datasets is <100 and the amino acid sequence length of target proteins is <1500. The vast majority of *pK_d_* in the Davis dataset are concentrated at 5, which corresponds to an extremely low affinity. The KIBA *scores* in the KIBA dataset are similarly concentrated in the middle part with a normal distribution.

### 4.2. Evaluation Metrics

Modeling DTA prediction as a regression task, we evaluated the performance of algorithms with the *MSE*, *CI*, and $r_{m}^{2}$.

*MSE* can measure the gap between the predicted affinity value and the actual affinity value.

The calculation formula for *CI* is as follows, where *p_i_* represents the predicted value of the sample with a larger affinity value *y_i_*, *p_j_* is the predicted value of the sample with a smaller affinity value *y_j_*, and *Z* is the normalization constant that equals the number of data pairs with different actual affinity values.
(3)$$CI=\frac{1}{Z}\sum_{y_{i}>y_{j}}h(p_{i}-p_{j})$$

*h*(.) is a piecewise function calculated as fellows.
(4)$$h(x)=\left\{\begin{array}{c} \;1\;,\\ 0.5,\\ \;0\;, \end{array}\begin{array}{c} \;x>0\\ \;x=0\\ \;x<0 \end{array}\right.$$

*CI* can evaluate the ranking ability of the algorithm, that is, whether the order of predicted drug-target affinity is consistent with the true order. The value of *CI* ranges from 0 to 1, and if *CI* > 0.5, it indicates that the algorithm performs well.

The calculation formula of $r_{m}^{2}$ is as follows, where *r*^2^ and $r_{0}^{2}$ are the square correlation coefficients with and without intercepts.
(5)$$r_{m}^{2}=r^{2}*(1-\sqrt{r^{2}-r_{0}^{2}})$$

$r_{m}^{2}$ evaluates the external prediction performance of the quantitative-structure activity relationship (QSAR), and $r_{m}^{2}$ > 0.5 represents that the performance of the algorithm is acceptable.

### 4.3. Proposed Algorithm Architecture

The proposed algorithm GRA-DTA includes three modules: drug molecular encoder, target protein encoder, and DTA prediction modules. We represent drug molecules with graphs and target proteins with one-hot encoding of amino acid sequences. Initially, target representations are learned via a BiGRU and soft attention-based target protein encoder. Ultimately, drug representations are learned via a GraphSAGE-based drug molecular encoder. Subsequently, drug and target representations are merged by ANN and fed into the fully connected layers to make DTA predictions. The overall architecture of GRA-DTA is shown in Figure 7.

### 4.4. Target Protein Representation

For targets in the benchmark datasets, as mentioned above, the amino acid sequences have been obtained, which are sequences of ASCII characters composed of 25 combinations of letters, where each letter represents a specific type of amino acid. We map these letters to integers from 0 to 24 and obtain the feature matrix $X^{t}\in R^{\;L\times C^{t}}$ of the target protein with one-hot encoding, where *L* is the length of the target sequence and $C^{t}$ = 25 represents the dimension of the amino acid features.

To maintain the integrity of the majority of sequence features while ensuring processing efficiency, according to the statistical results of the dataset mentioned above, the target protein sequence length is fixed to *L* = 1000. For target protein sequences that exceed 1000, some parts are truncated, and those that are insufficient are padded with 0.

### 4.5. Drug Molecular Representation

For drugs in the benchmark datasets, as mentioned above, the SMILES have been obtained, and RDKit was used to construct a drug molecular graph. With the atoms as the nodes and the chemical bonds as the edges, a 2D graph *G^d^*= (*V*, *E*) of the drug molecule was established in which $V={\left\{v_{i}\right\}}_{i=1}^{N}$ is the set of atoms, $E={\left\{e_{j}\right\}}_{j=1}^{M}$ is the set of edges, *N* is the number of atoms, and *M* represents the number of chemical bonds. The digitized representation of a drug molecular graph is represented by the edge index $EI\in R^{\;2\times M}$ and the node feature matrix $X^{d}\in R^{\;N\times C^{d}}$, where $C^{d}$ = 78 is the dimension of node features.

The node features adopt the atomic characteristics adapted from DeepChem [40], which include the atomic category (44 categories), degree of the node (0–10), hydrogen atom quantity connected to the atom (0–10), valence of the atom (0–10), and whether the atom has aromaticity (true or false). We use one-hot encoding to encode these categories of features into a $C^{d}$ = 78-dimensional binary feature vector.

### 4.6. Target Protein Encoder Based on BiGRU and Soft Attention

The target feature matrix *X^t^* passes through an embedding layer to obtain the embedding representation $X^{t}\in R^{\;L\times D^{t}}$, where *D^t^* is the amino acid embedding dimension. Subsequently, BiGRU is used to extract the features of the target protein sequence as shown in Figure 7A.

The target protein sequence is regarded as a time series $X^{t}=\{x_{1},\;x_{2}\ldots \;x_{L}\},t=1\;\ldots \;L$. Supposing *x_t_* is the input at time step *t*, *h_t_* is the hidden state of GRU at time *t*. The hidden state at time *t* is related to the hidden state at time *t* − 1, and GRU controls the flow of information through a gating mechanism. The calculation formula of update gate *z_t_* is as follows:(6)$$z_{t}=\sigma (x_{t}W_{z}+h_{t-1}U_{z})$$

The calculation formula of reset gate *r_t_* is as follows:(7)$$r_{t}=\sigma (x_{t}W_{r}+h_{t-1}U_{r})$$
where $W_{z}$, $U_{z}$, $W_{r}$, and $U_{r}$ are the learnable weight matrices. The weight matrix is a key component of all kinds of neural networks, $W_{z}$ and $U_{z}$ determine the degree of influence of input $x_{t}$ and $h_{t-1}$ on output $z_{t}$. Similarly, $W_{r}$ and $U_{r}$ determine the extent to which the input $x_{t}$ and $h_{t-1}$ influence the output $r_{t}$. They are learned during the training process by the back-propagation algorithm. *σ*(.) is the Sigmod activation function, so the values of *z_t_*, *r_t_* are between 0–1. The candidate state ${\tilde{h}}_{t}$ is calculated as follows:(8)$${\tilde{h}}_{t}=\tanh(x_{t}W_{x}+(r_{t}⊙h_{t-1})U_{x})$$
where *r_t_* controls the proportion of information acquired from the historical state *h_t_*_−1_ to the candidate state *h_t_*, *W_x_* is the learnable weight matrix, tanh(.) is the tanh activation function, and ⊙ is element-wise multiplication. The hidden state at time *t* is calculated as follows:(9)$$h_{t}=(1-z_{t})⊙h_{t-1}+z_{t}⊙{\tilde{h}}_{t}$$
where *z_t_* controls the proportion of currently hidden state *h_t_* to obtain information from historical state *h_t_*_−1_ and candidate state ${\tilde{h}}_{t}$.

The information flow in GRU is unidirectional, moving solely from the previous context to the current, while the characteristics of a segment within a protein sequence are not exclusively linked to the preceding context. Hence, we consider introducing BiGRU, which comprises two GRUs in opposite directions. At each time step, the input simultaneously integrates the hidden states of these two GRUs, and the output is jointly determined by both unidirectional GRUs. The specific calculation is as follows:(10)$$\begin{array}{l} \vec{h_{t}}=\mathrm{GRU}(x_{t},{\vec{h}}_{t-1})\\ \overset{\leftarrow }{h_{t}}=\mathrm{GRU}(x_{t},{\overset{\leftarrow }{h}}_{t-1})\\ h_{t}=\vec{h_{t}}||\overset{\leftarrow }{h_{t}} \end{array}$$
where ${\vec{h}}_{t}$ and ${\overset{\leftarrow }{h}}_{t}$ are the forward and backward hidden states, respectively, || represents the concatenation operation, and $h_{t}$ is the final output at time *t*.

Through BiGRU, the hidden state of the target protein sequence is obtained and denoted as $H^{t}=\{h_{1},h_{2}\;\ldots \;h_{L}\}$, $h^{t}\in R^{2D^{t}}$. We introduce a soft attention mechanism to focus on the key information related to DTA in the long sequence of target proteins, as shown in Figure 7B. The attention weight vector *α_i_* of the *i*-th hidden state is calculated as follows:(11)$$\alpha_{i}=\mathrm{softmax}(s(h_{i}))=\frac{\exp(s(h_{i}))}{\sum_{j=1}^{L}\exp(s(h_{j}))}$$
where *s*(.) is the attention score function and is calculated as follows:(12)$$s(h_{i})=U_{a}.\tanh(W_{a}h_{i})$$
where *U_a_*, and *W_a_* are learnable weights matrices and tanh(.) is the tanh function. Finally, the output $att(H^{t})$ of the attention layer is obtained by the weighted sum of inputs according to attention weights:(13)$$att(H^{t})=\sum_{i=1}^{L}\alpha_{i}h_{i}$$

After that, we down-sample the $att(H^{t})$ by a linear layer, obtaining the representation of the target protein $Y^{t}\in R^{D}$. *D* denotes the vector dimension.

### 4.7. Drug Molecular Encoder Based on GraphSAGE

For the drug molecule graph *G*, the initial node feature is $X^{d}=\{x_{1},\;x_{2}\ldots \;x_{N}\},v=1\ldots \;N$. We use GraphSAGE, whose efficacy in learning molecular representations has proven to learn the node embeddings [41].

The core idea of GraphSAGE is to sample and aggregate neighborhoods. Assuming there is a *K*-layer network, for the central node *v*, the initial node embedding $h_{v}^{0}=x_{v}$, in each layer, a fixed size of neighbors *Z* was sampled (if the number of neighbor nodes is less than *Z*, repeated sampling is performed), then the embedding of *v* was updated by aggregating information from its neighboring nodes. The details of GraphSAGE are shown in Figure 7C.

The aggregation function is Equation (14), where $h_{N(v)}^{k}\in R^{D^{d}}$ denotes the embedding of the neighbors of *v* in the *k*-th layer, which are obtained by mean aggregation:(14)$$\begin{array}{l} h_{N(v)}^{k}\leftarrow \mathrm{mean}(\sigma (W_{pool}^{k}h_{u_{i}}^{k-1}+b))\\ \forall u_{i}\in N(v) \end{array}$$

$h_{u_{i}}^{k-1}$ is the embedding of neighbors of *v* in the *k*−1-th layer, *N*(*v*) represents the set of neighbors of *v*, $W_{pool}^{k}$ is the learnable weight matrix, *b* is the bias term, and *σ*(.) is the ReLU activation function. Subsequently, the node embedding of *v* in the *k*-th layer, represented by $h_{v}^{k}$, are updated according to Equation (15):(15)$$h_{v}^{k}\leftarrow \sigma (W_{u}^{k}.(h_{v}^{k-1}||h_{N(v)}^{k}))$$
where $W_{u}^{k}$ is the learnable weight matrix and || represents the concatenation operation.

We also conduct a batch normalization following each GraphSAGE layer activated by a ReLU function to alleviate the vanishing or exploding gradient.

After several GraphSAGE layers, we choose global max pooling to aggregate the learned node embeddings to learn the most significant features in the drug molecular graph and pass it through a linear layer with the ReLU activation function to obtain the final drug representation $Y^{d}\in R^{D}$. *D* denotes the vector dimension.

### 4.8. DTA Prediction Based on ANN

After obtaining $Y_{i}^{d}$ of drug *i* and $Y_{j}^{t}$ of target *j*, prior research employed a simple concatenation to obtain drug-target pair representation. However, the importance of different parts of the feature representations varies in distinct drug-target interactions. Drawing inspiration from Cheng et al. [42], we introduce the ANN to enhance the representation of drug-target pairs by fusing $Y_{i}^{d}$ and $Y_{j}^{t}$, as shown in Figure 7D. This module is capable of capturing varying attention strengths associated with dimensions of drug-target pairs.

The representation of drug-target pair $V_{ij}\in R^{D}$ is characterized as:(16)$$V_{ij}=\alpha_{ij}⊙(Y_{i}^{d}⊙Y_{j}^{t})$$
where *α_ij_* is an attention vector that can capture the importance of different dimensions. The attention coefficient *a_i,j,k_* of the *k*-th (*k* = 1, 2 … *D*) dimension is calculated as follows:(17)$$a_{i,j,k}=\mathrm{softmax}({\hat{a}}_{i,j,k})=\frac{\exp({\hat{a}}_{i,j,k})}{\sum_{m=1}^{D}\exp(a_{i,j,m})}$$
where ${\hat{a}}_{i,j,k}$ is the attention score, and is calculated by:(18)$${\hat{a}}_{i,j,k}=U_{a}\cdot \sigma (W_{a}(Y_{i}^{d}||Y_{j}^{t}))$$
where *U_a_*, *W_a_* are learnable weight vectors, *σ*(.) is the ReLU activation function.

Finally, the drug representation $Y_{i}^{d}$, target representation $Y_{j}^{t}$, and drug-target pair representation $V_{ij}$ are concatenated and then input into two-layer fully connected networks, each followed by a dropout to prevent overfitting and a ReLU activation function. After that, a continuous value of predicted affinity *y_i_* is obtained through a final fully connected layer.

*MSE* is adopted as the loss function:(19)$$MSE=\frac{1}{n}\sum_{i=1}^{n}{\left(p_{i}-y_{i}\right)}^{2}$$
where *p_i_* denotes the actual affinity value and *n* is the number of drug-target pairs in the training set.

## 5. Conclusions

In this article, we proposed a novel algorithm called GRA-DTA to predict DTA. The drug molecules are represented as a graph and GraphSAGE is used to learn drug representations. The target protein sequence representations are learned with soft attention-based BiGRU to effectively focus on the global features of the long sequence. Finally, the drug and target representations are fused by an ANN to learn the representations of drug-target pairs and feed them into fully connected layers to obtain prediction results. Experimental results on benchmark datasets prove that GRA-DTA exhibits commendable performance and outperform five baseline algorithms in certain metrics. Concurrently, we also evaluate GRA-DTA under three cold start scenarios, which proves its superior generalization capability in comparison to the baseline algorithms on the KIBA dataset, as well as the cold target and cold drug-target scenarios of the Davis dataset. Ultimately, a case study focusing on COVID-19 target 3CLpro demonstrates the potential of GRA-DTA for real-world drug repurposing applications.

In future work, we aim to incorporate drug fingerprint [43] features to enhance drug information and bolster drug discrimination. Furthermore, with the advent of the robust protein structure prediction tool AlphaFold2, we can construct a protein graph utilizing the predicted 3D structure, which would leverage the topological features of proteins to enhance DTA prediction.

## Figures and Tables

**Figure 1 ijms-25-05126-f001:**
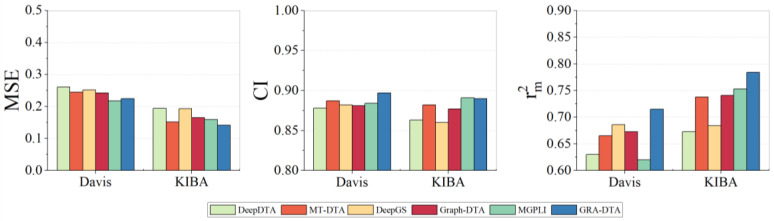
Performance comparison of GRA-DTA and baseline algorithms.

**Figure 2 ijms-25-05126-f002:**
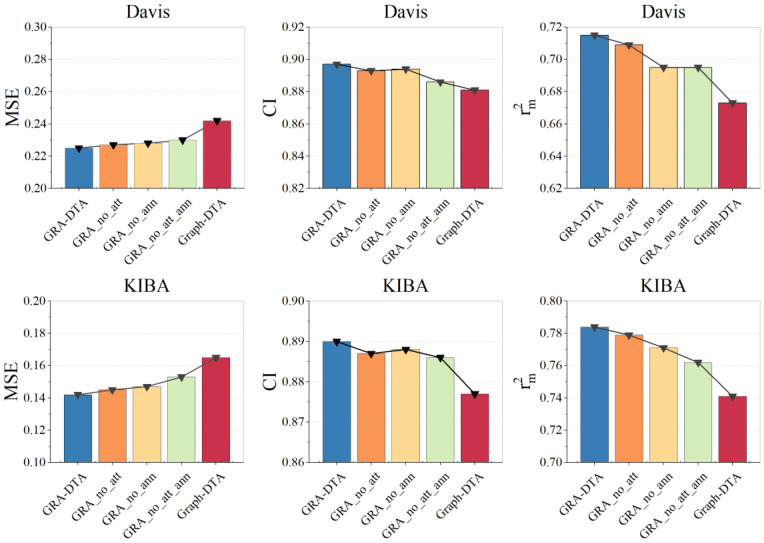
Results of ablation experiments for GRA-DTA.

**Figure 3 ijms-25-05126-f003:**
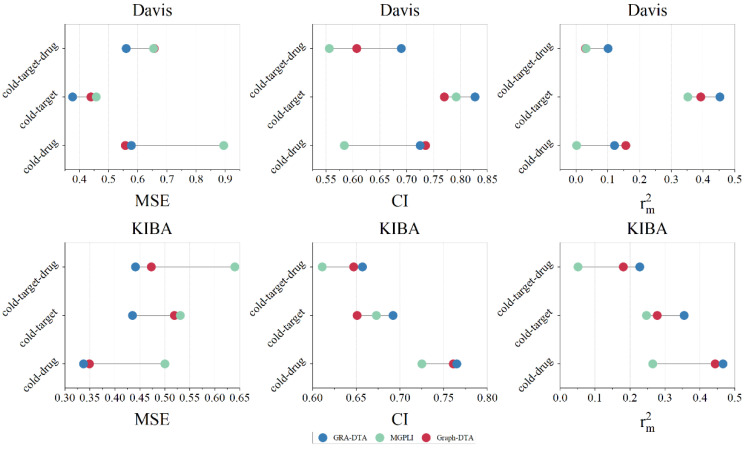
Performance comparison of GRA-DTA and baseline algorithms on cold start scenarios.

**Figure 4 ijms-25-05126-f004:**
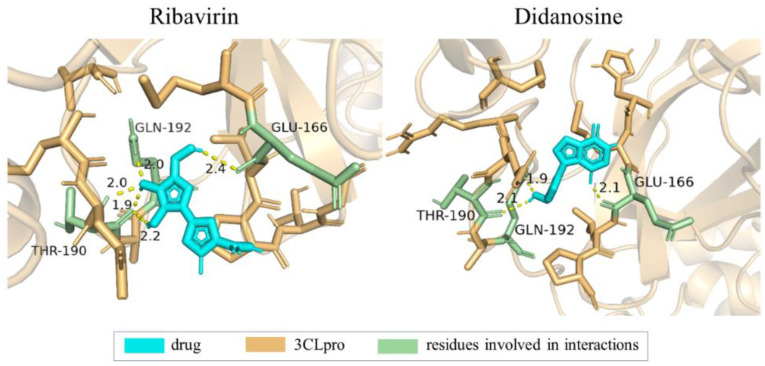
Three-dimensional pose of ligand-protein binding state between drugs ((**left**): Ribavirin with a binding energy of—7.35 kcal/mol; (**right**): Didanosine with a binding energy of—5.08 kcal/mol) and target 3CLpro, where cyan part is drug molecule and yellow part is target protein, in which residues that interact with the drug are represented in green. Yellow dashed line is hydrogen bond between residues and drug atoms, and number in black represents bond length (Å).

**Figure 5 ijms-25-05126-f005:**
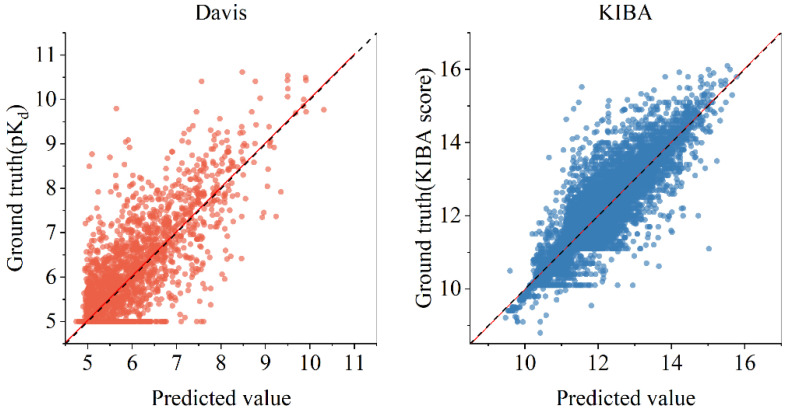
Relationship between prediction value of GRA-DTA and ground truth. For Davis dataset, linear fit result (red solid line) has slope of 0.998 and intercept of 0.036. For KIBA dataset, linear regression result (red solid line) has slope of 1.005 and intercept of −0.07. the black dotted line is *y* = *x* representing perfect prediction.

**Figure 6 ijms-25-05126-f006:**
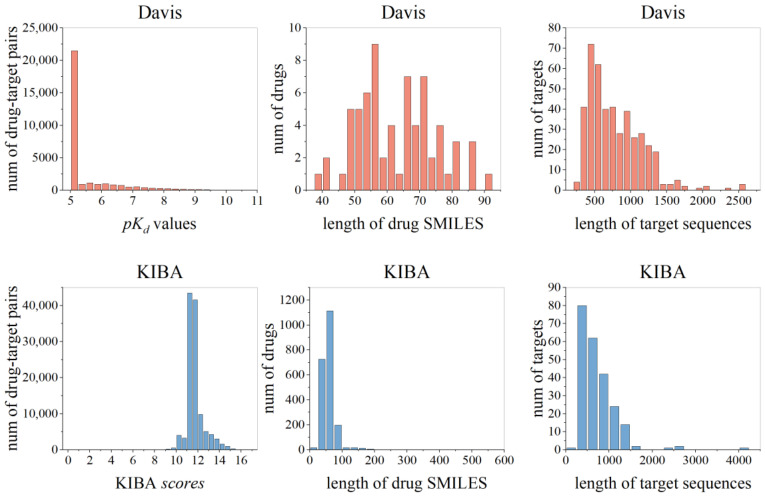
Distribution ranges of affinity values, length of drug SMILES, and length of target sequences in Davis and KIBA datasets.

**Figure 7 ijms-25-05126-f007:**
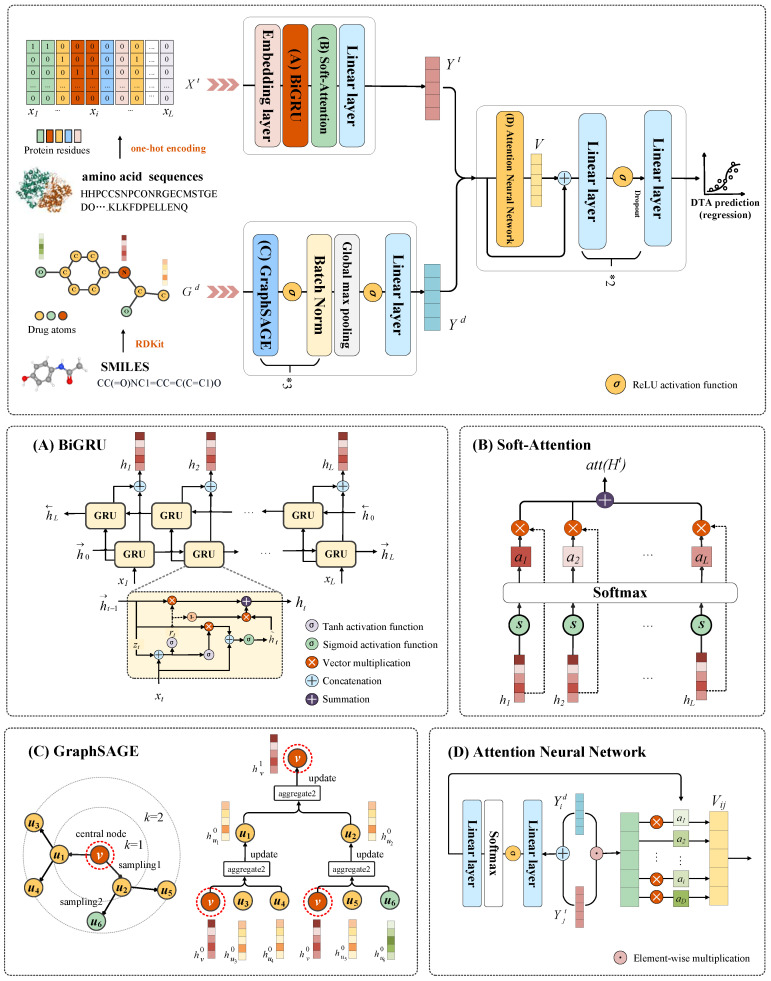
Overall architecture of GRA-DTA. (**A**) Structure details of BiGRU, (**B**) process details of soft-attention mechanism, (**C**) sample and aggregation details of GraphSAGE, (**D**) process details of ANN.

**Table 1 ijms-25-05126-t001:** Hyperparameter settings of GRA-DTA.

Hyperparameter	Value
Epochs	600
Batch size	128 (Davis), 256 (KIBA)
Learning rate	2 × 10^−4^
Optimizer	Adam
Dropout rate	0.2
Layer of BiGRU	2
Layer of GraphSAGE	3
Node embedding dimension of drug molecular encoder	{32, 64, **128**, 256}
Hidden-state dimension of target protein encoder	{32, 64, **128**, 256}
Dimension of drug and target representations	{64, **128**, 256}

Bold: optimal parameters for the best performance.

**Table 2 ijms-25-05126-t002:** Comparative results of GRA-DTA and baseline algorithms.

Dataset	Algorithm	MSE ↓	CI↑	$r_{m}^{2}$ ↑
Davis	DeepDTA	0.261	0.878	0.630
	MT-DTA	0.245	0.887	0.665
	DeepGS	0.252	0.882	0.686
	Graph-DTA	0.242	0.881	0.673
	MGPLI	**0.218**	0.884	0.620
	GRA-DTA	0.225	**0.897**	**0.715**
KIBA	DeepDTA	0.194	0.863	0.673
	MT-DTA	0.152	0.882	0.738
	DeepGS	0.193	0.860	0.684
	Graph-DTA	0.165	0.877	0.741
	MGPLI	0.159	**0.891**	0.753
	GRA-DTA	**0.142**	0.890	**0.784**

Bold: best value; underlined: second-best value; ↑: larger values representing better performance; ↓: smaller values representing better performance.

**Table 3 ijms-25-05126-t003:** Results of ablation experiments for GRA-DTA.

Dataset	Variant	MSE ↓	CI ↑	$r_{m}^{2}$ ↑
Davis	GRA	**0.225**	**0.897**	**0.715**
	GRA_no_att	0.227	0.893	0.709
	GRA_no_ann	0.228	0.894	0.695
	GRA_no_att_ann	0.230	0.886	0.695
	Graph-DTA	0.242	0.881	0.673
KIBA	GRA	**0.142**	**0.890**	**0.784**
	GRA_no_att	0.145	0.887	0.779
	GRA_no_ann	0.147	0.888	0.771
	GRA_no_att_ann	0.153	0.886	0.762
	Graph-DTA	0.165	0.877	0.741

Bold: best value; underlined: second-best value; ↑: larger values representing better performance; ↓: smaller values representing better performance.

**Table 4 ijms-25-05126-t004:** Performance comparison of GRA-DTA and baseline algorithms on cold start scenarios.

Scenario		Davis			KIBA		
	Algorithm	MSE ↓	CI ↑	$r_{m}^{2}$ ↑	MSE ↓	CI ↑	$r_{m}^{2}$ ↑
cold-drug	GraphDTA	**0.557**	**0.735**	**0.157**	0.349	0.761	0.444
	MGPLI	0.895	0.584	0.002	0.50	0.725	0.265
	GRA-DTA	0.578	0.725	0.121	**0.337**	**0.765**	**0.466**
cold-target	GraphDTA	0.439	0.770	0.393	0.519	0.651	0.278
	MGPLI	0.457	0.792	0.352	0.531	0.673	0.247
	GRA-DTA	**0.376**	**0.827**	**0.453**	**0.435**	**0.692**	**0.355**
cold-target-drug	GraphDTA	0.656	0.607	0.030	0.473	0.647	0.181
	MGPLI	0.654	0.556	0.032	0.640	0.611	0.051
	GRA-DTA	**0.560**	**0.690**	**0.101**	**0.441**	**0.657**	**0.228**

Bold: best value; underlined: second-best value; ↑: larger values representing better performance; ↓: smaller values representing better performance.

**Table 5 ijms-25-05126-t005:** Ranks of the top 10 antiviral drugs and 3 unrelated drugs predicted by GRA-DTA.

Rank	Drug	Structure	PubMed ID	Rank	Drug	Structure	PubMed ID
1	Ribavirin	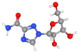	32227493 [28];34991982 [29];34916812 [30]	8	Acyclovir	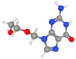	36101854 [31]
2	Didanosine	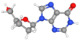	35785294 [32]	9	Entecavir	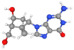	-
3	Zalcitabine	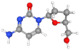	-	10	Abacavir	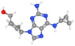	33916747 [33]
4	Peramivir	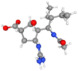	-	67	Artemisinin	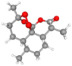	-
5	Etravirine	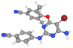	33168456 [34];35409412 [35]	79	Penicillin	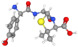	-
6	Taribavirin	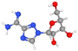	-	87	Aspirin	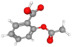	-
7	Methisazone	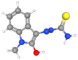	32278693 [36]				

**Table 6 ijms-25-05126-t006:** Summary of Davis and KIBA datasets.

	Davis	KIBA
No. of drugs	68	2111
No. of proteins	442	229
No. of binding affinities	30,056	118,254
Maximum length of drugs	103	590
Maximum length of proteins	2549	4128
The average length of drugs	64	58
The average length of proteins	788	728
Affinity Measures	*pK_d_*	KIBA *score*
Range of affinities	5.0~10.8	0.0~17.2

## Data Availability

The required data are available on GitHub (https://github.com/thinng/GraphDTA, accessed on 10 October 2023).

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
