# Peer review of "Prediction of Drug-Target Affinity Using Attention Neural Network"

_ijms, 2024, doi:10.3390/ijms25105126_

Round 1
Reviewer 1 Report
Comments and Suggestions for Authors
The manuscript “Prediction of Drug-Target Affinity Using Attention Neural Network” presents a novel algorithm for drug-target affinity analysis. The manuscript is very interesting and well written. But some issues need to be improved prior to publication. My suggestion is a major revision according to the following comments:
The abstract should include some specific numerical results that confirm the advantages of the developed algorithm.
Equation 1. Please improve the writing of the exponent in the denominator of the fraction.
Section 3.1. should be part of the Materials and Methods section.
The materials and method section should include information about the dimensions of the data matrices used for modeling. Did the authors divide the data set into calibration and independent prediction? Please explain.
The discussion section should include some comparisons with the available literature.
Reviewer 2 Report
Comments and Suggestions for Authors
In the artile titled 'Prediction of Drug-Target Affinity Using Attention Neural Network' by Tang et al., a novel deep learning algorithm called GRA-DTA is proposed. It combines Bidirectional Gated Recurrent Unit with a soft attention mechanism to learn target representations. Additionally, Graph Sample and Aggregate is used to learn drug representations. The model predicts drug-target pairs with an attention neural network that merges drug and target representations. Finally, experimental results on the benchmark datasets KIBA and Davis show that GRA-DTA achieves better performance compared with all other state-of-the-art DTA prediction algorithms. The article is acceptable for publication in IJMS after minor revision.
Some comments follow.
p. 1 l. 33 It should be clear that MD stands for Molecular Dynamics, which is used universally.
p.1 l.70 Targets are represented as residue squences, however the affinity does not correlate with the entire protein structure. You can perform an affinity test of a synthetic protein consisting in the 50 % of the natural one and obtain a 99 % of its affinity. It is a local phenomenon depending of the structure of the pocket. Authors should comment these ideas.
p. 3 l. 98 Authors call GRA-DTA their algorithm. What GRA stands for?
p. 3 l.127 Eq. 1 1e9 should be changed by 10^9
p. 5 Fig 1 ReLu activation function is not defined
p. 6 l.200 Authors should explain what are the weight matrices
p. 7 l.215 Eq. 7. This is where the authors should say that the operator || means concatenation.
p. 9 In the calculation of CI, how Z is established? Does not woluld be better to use a well-established paramenter with the same purpose as is the Spearman correlation coefficient?
p. 9 l.312 'et al' should be deleted
Tables 3 and 4. An horizontal line to separate the results for each dataset would help the visualization of the table.
p.10 l.323 'Figure 4' should be 'Figure 3'.
p.11 l.343 The red solid line delineates the linear fit to the scatter points, and the black dashed line is y = x representing perfect prediction. However, it is strange, because the points do no seem symmetrically distributed. Probably the authors have performed a correlation forcing the constant=0?
p.13 384-389 Authors point out that the enhancement of MSE is challenging on the Davis dataset. Could it be due to that the main part of affinities in Davis are so poor?
References are not in the journal's format.
References [10], [14], [24] and [25] are online archives, but specific URLs or DOI numbers are missing.
The feedback provided for each of the preceding points should be incorporated into the revised version of the paper, ensuring that the comments are not solely addressed in response to this specific reviewer but are also evident in the updated manuscript.
Comments on the Quality of English Language
Grammar needs to be checked more carefully.
Round 2
Reviewer 1 Report
Comments and Suggestions for Authors
The authors put in a significant effort and answered most of my comments. I think they improved the manuscript. Therefore, manuscript can be accepted for publication.